# Predicting medical device failure: a promise to reduce healthcare facilities cost through smart healthcare management



Noorul Husna Abd Rahman[1,2], Muhammad Hazim Mohamad Zaki[1], Khairunnisa Hasikin[1,3], Nasrul Anuar Abd Razak[1], Ayman Khaleel Ibrahim[4] and Khin Wee Lai[1]

[1] Department of Biomedical Engineering, Universiti Malaya, Lembah Pantai, Wilayah Persekutuan Kuala Lumpur, Malaysia
[2] Engineering Services Division, Ministry of Health, Putrajaya, Wilayah Persekutuan Putrajaya, Malaysia
[3] Center of Intelligent Systems for Emerging Technology (CISET), Faculty of Engineering, Universiti Malaya, Lembah Pantai, Wilayah Persekutuan Kuala Lumpur, Malaysia
[4] Faculty of Computing and Informatics, Universiti Malaysia Sabah, Kota Kinabalu, Sabah, Malaysia

Corresponding author
Khairunnisa Hasikin,
khairunnisa@um.edu.my

## ABSTRACT

**Background:** The advancement of biomedical research generates myriad healthcare-relevant data, including medical records and medical device maintenance information. The COVID-19 pandemic significantly affects the global mortality rate, creating an enormous demand for medical devices. As information technology has advanced, the concept of intelligent healthcare has steadily gained prominence. Smart healthcare utilises a new generation of information technologies, such as the Internet of Things (loT), big data, cloud computing, and artificial intelligence, to completely transform the traditional medical system. With the intention of presenting the concept of smart healthcare, a predictive model is proposed to predict medical device failure for intelligent management of healthcare services.

**Methods:** Present healthcare device management can be improved by proposing a predictive machine learning model that prognosticates the tendency of medical device failures toward smart healthcare. The predictive model is developed based on 8,294 critical medical devices from 44 different types of equipment extracted from 15 healthcare facilities in Malaysia. The model classifies the device into three classes; (i) class 1, where the device is unlikely to fail within the first 3 years of purchase, (ii) class 2, where the device is likely to fail within 3 years from purchase date, and (iii) class 3 where the device is likely to fail more than 3 years after purchase. The goal is to establish a precise maintenance schedule and reduce maintenance and resource costs based on the time to the first failure event. A machine learning and deep learning technique were compared, and the best robust model for smart healthcare was proposed.

**Results:** This study compares five algorithms in machine learning and three optimizers in deep learning techniques. The best optimized predictive model is based on ensemble classifier and SGDM optimizer, respectively. An ensemble classifier model produces 77.90%, 87.60%, and 75.39% for accuracy, specificity, and precision compared to 70.30%, 83.71%, and 67.15% for deep learning models. The ensemble classifier model improves to 79.50%, 88.36%, and 77.43% for accuracy, specificity, and precision after significant features are identified. The result concludes although machine learning has better accuracy than deep learning, more training time is

required, which is 11.49 min instead of 1 min 5 s when deep learning is applied. The model accuracy shall be improved by introducing unstructured data from maintenance notes and is considered the author's future work because dealing with text data is time-consuming. The proposed model has proven to improve the devices' maintenance strategy with a Malaysian Ringgit (MYR) cost reduction of approximately MYR 326,330.88 per year. Therefore, the maintenance cost would drastically decrease if this smart predictive model is included in the healthcare management system.

## INTRODUCTION

Medical devices are used for the diagnosis and treatment of disease, as well as the rehabilitation of patients during post-treatments (*Aridi et al., 2016*). It can be used independently or in conjunction with any accessory, consumable, or other pieces of medical device. The reliability, maintainability, availability, and safety of medical devices are the ultimate goal in maintenance strategy, where medical devices should not fail frequently and must be repaired promptly when the failures are detected. Numerous investigations have associated medical device failures with severe patient injuries and deaths (*Dhillon, 2011*; *Mahfoud, Abdellah & El Biyaali, 2018*; *Palmer, 2010*; *Verbano & Turra, 2010*). The world health organization (WHO) estimates that 50% to 80% of equipment is non-functional due to the lack of maintenance culture, competency, and a tendency to focus on corrective maintenance rather than preventative maintenance (*Kutor, Agede & Ali, 2017*). Ineffective medical device maintenance causes crucial equipment downtime, diminished device performance, monetary waste, and depletion of resources (*Bahreini, Doshmangir & Imani, 2019*). Older technological devices require more attention due to a lack of service or user manuals and manufacturer guidance (*Engineering Services Division Ministry of Health, 2018*; *Sezdi, 2016*). Meanwhile, among the leading causes of downtime or equipment failures are inadequate storage and transportation, initial failure, inappropriate handling (damage during use), inadequate maintenance, use of non-genuine spare parts or refurbished spare parts, environmental stress, random failure, improper repair technique, and wear-out failures (*Kutor, Agede & Ali, 2017*).

Medical devices range from basic tongue depressors to complex radiation systems with over 10,000 different types, and 1.5 million unique medical devices are recorded globally. Medical device expenditure climbed from USD 145 billion in 1998 to USD 220 billion in 2006, which indicates an annual growth rate of more than 10% (*World Health Organization, 2011a*). Medical devices maintenance market projection demonstrates that the medical devices maintenance market is expected to grow at a compound annual growth rate (CAGR) of 10.4% to USD 74.2 billion by 2026, which will increase drastically from USD 45.2 billion in the year 2021 (*Markets and Markets, 2021*). The increased

emphasis on early diagnosis, the rising number of diagnostic imaging procedures, the presence of a large number of original equipment manufacturers (OEMs), and strategic partnerships and collaborations between service providers and end-users are all propelling the medical devices maintenance market growth. Implementing smart management technology, such as asset management solutions, incurs hefty installation and ongoing maintenance expenses. The installation of modern medical devices is accompanied by a service contract requiring annual payments of approximately 12% of the cost of medical devices. The total maintenance cost is usually more significant throughout the device's lifespan than the device's cost (*General, 2021*; *Markets and Markets, 2021*). A similar scenario is apprehended in most developing countries, especially in Malaysia, where the Auditor's General Report from 2016 to 2020 indicates the medical devices asset values in Ministry of Health, Malaysia reached RM7.775 billion with RM2.118 billion of medical device maintenance costs is fully utilized within the said years. Malaysian government hospitals have a large quantity of outdated medical devices, and the maintenance cost is rising with their age where with 40.7% of active medical device has less than 10 years in service, 39.7% for 11–20 years, 16.7% for 21–30 years and 2.9% more than 30 years and still in service (*General, 2021*).

Equipment failure and equipment uptime are critical for efficient healthcare delivery in any country. The necessity to reduce maintenance expenses while prolonging the device's lifecycle drives the development of an efficient medical maintenance plan (*Mahfoud, El Barkany & Biyaali, 2017*). Due to the large number of medical devices used in healthcare facilities, various maintenance techniques have been applied to ensure reliability. Prioritization of maintenance procedures has been advocated to assure uptime and reduce the cost of maintenance or replacement. Budget limits for maintenance and replacement are essential to address and always be a concern. Maintenance and replacement costs are reduced by categorizing medical equipment according to their criticality (*Hutagalung & Hasibuan, 2019*). The necessity of prioritization is also explained in *Hilmi et al. (2021)* and *Mahfoud, Abdellah & El Biyaali (2018)*, where most medical equipment has a complicated system in repair and a significant number of connected components, which directly impacts patients and requires prioritization for planned maintenance to avoid failures (*Mahfoud, Abdellah & El Biyaali, 2018*). A study on smart prioritization programs has been explored by *Zamzam et al. (2021)* based on preventive, corrective, and replacement programs. Three robust models were developed for effective smart management of healthcare facilities into low, medium, and high categories. Efficient medical device maintenance ensures a longer lifespan, functionality, and reliability by relying on scientific and engineering principles, biomedical engineering education, previous maintenance history and experience, manufacturer recommendations, expert suggestions, and the obligation to comply with country regulatory requirements (*Khalaf et al., 2010*). Maintenance program effectiveness and efficiency are evaluated using various methodologies to minimize or eliminate hazards, including maintenance history data, physical inspection, and failure analysis. This could potentially be done by leveraging big data and smart management systems for efficient healthcare delivery services.

Meanwhile, in Malaysia's healthcare facilities, preventive maintenance is performed at a set time or interval throughout the year, as recommended by the manufacturer or as scheduled, and corrective maintenance is conducted after failure where the repair work is implemented (*Coban et al., 2018*). Clinical engineers are responsible for calibration, maintenance, repair, user training, and decommissioning of medical devices by applying engineering and managerial skills (*World Health Organization, 2011b*). Another aspect that influences the dependability and failures of medical equipment is the level of knowledge of users and biomedical staff. There is a challenge to employ an expert or well-trained worker in biomedical engineering; thus, outsourcing or implementing a service contract with specialized contractors or vendors is an option (*Mahfoud, Abdellah & El Biyaali, 2018*). Due to the same limitation, Malaysian Government hospitals have adopted a maintenance service contract for medical devices and launched a privatization program with Concession Company in 1997. Other elements that influence the performance of medical equipment include calibration work and electrical safety testing, planned maintenance, and competence. Calibration is a task to verify the accuracy, and an electrical safety test is performed to ensure the patient is not at risk of electrical injury or leakage. According to the results of an investigation test conducted on six high-risk medical devices, 58% of the devices failed the performance test, which is greater than prior studies, which found that 21% and 26% of the devices failed the performance test, respectively (*Altayyar et al., 2018*). Furthermore, when measured in healthcare facilities, around 9% of infusion pumps and 12.6% of dialysis machines fail to meet electrical safety regulations (*Gurbeta Pokvic et al., 2017a*, *2017b*). Meanwhile, clinical chemistry analyzers and infusion pumps were the most commonly reported medical device affected by electrostatic discharge failures due to a current flow that caused a dielectric breakdown (*Kohani & Pecht, 2018*). As a result, electrical safety tests are required in Malaysia for scheduled maintenance or newly purchased medical devices, and calibration work is performed on specified types of medical devices as advised by the manufacturer.

Healthcare administrators and biomedical engineers are continually confronted with issues pertaining to the security of their facilities, the satisfaction of their employees, the quality enhancement and improvement of the standard grade of care provided to their patients, the expensive workflow, and the inefficiency of costly processes. Numerous times, improvements in information technology (IT) for the healthcare system are cited as potential enhancement techniques. Innovators and researchers are constantly working to develop new technologies that can aid corporate operations and improve the quality of activities. Today's healthcare facilities are confronted with an increase in the number of patients and heightened expectations for patient experiences and levels of satisfaction as an essential consequence of healthcare services. To fulfill these increasing demands, it is necessary to eliminate unnecessary stages and simplify the workflow. Artificial intelligence (AI)—assisted asset management is one of the primary components of an efficient work process. AI in healthcare applications and model development has emerged as a promising tool in providing a solution to humans while dealing with a crisis, especially during the COVID-19 pandemic. It is utilized during decision-making through feature learning (*Ghorbani et al., 2020*; *Jayatilake & Ganegoda, 2021*; *Kulathilake et al., 2021*). Pre-

processing, feature extraction, and classification are the three steps to be examined in machine learning (ML) techniques for healthcare applications (*Jayatilake & Ganegoda, 2021*). The execution time is lowered, and the classification accuracy is raised by implementing feature selection before classification. Principal component analysis (PCA) and genetic algorithms can be used to generate a feature set for classification (*Jayatilake & Ganegoda, 2021*).

PCA is a technique to avoid overfitting and is used for performance advancement and noise reduction. The PCA is applied with 16 principal components to obtain 95% of the original variance using a random forest algorithm in ML to detect Chagas disease (*Morais et al., 2022*). A genetic algorithm (GA) is a search heuristic that imitates the natural evolution process. It is frequently employed to find practical answers to optimization and search challenges (*Santra & Christy, 2012*). The GA for feature selection has also been proposed by *Ghorbani et al. (2020)* and *Santra & Christy (2012)* in the clustering technique. However, they are most commonly produced in supervised learning, where data class labels are known. The primary goal is to reduce the number of features utilized in classification while retaining acceptable classification accuracy. Prioritization, failure, and risk analysis are all employed in various applications of medical device reliability under the risk management area. Failure mode and effect analysis (FMEA) (*Arathy & Balasubramanian, 2020*), a mix of FMEA and Fuzzy (FFMEA) (*Jamshidi et al., 2015*), analytical hierarchy process (AHP) (*Hutagalung & Hasibuan, 2019*), and other methodologies are commonly used in risk management subjects. A study on medical device prioritization using a support vector machine is discussed in *Zamzam et al. (2021)*, where the technique outperforms preventive maintenance and replacement prioritization with 99.42% and 99.80% accuracy, respectively. In addition, K-nearest neighbour had the best accuracy of 98.9% in corrective maintenance prioritization. However, the study shall be enhanced to critical medical devices at more extensive facilities and broader clinical services such as hospitals. ML application in healthcare services is used in three research studies for medical devices performance prediction as described in *Badnjević et al. (2019)*, *Hrvat et al. (2020)* and *Spahić et al. (2020)*. However, these studies are limited to only one type of medical device to evaluate medical devices' performance.

In a study by *Ngabo et al. (2021)*, the ML model was developed based on the COVID-19 patient data to predict the patients' survival rate with the kNN algorithm, and Decision Trees attained the highest accuracy of 99.30%. Meanwhile, in a study by *Iwendi et al. (2020)*, a Boosted Random Forest algorithm attained an accuracy of 94% in predicting the severity of COVID-19 cases using patients' data and symptoms. Recently, the deep learning (DL) technique evolved as a sophisticated tool primarily used in medical imaging, text data, time series, and various image applications (*Ravikumar et al., 2022*). It is well known as a subdivision of ML that consists of different processing layers equipped with inputs, hidden, and output layers. Furthermore, researchers have explored the possibility of using long short term memory (LSTM) and deep reinforcement learning to predict losses and cures of patients' symptoms in the following few days after contracting the disease (*Kumar et al., 2021*). Besides, the admission and mortality of COVID-19 patients are predicted using an interpretable DL model with an area under curve (AUC) of 88.3%

(*Nazir & Ampadu, 2022*). The application of AI is also recently used for privacy and security issues (*Hameed et al., 2021*) and is widely used in smart and mobile healthcare (*Yamakoshi, Rolfe & Yamakoshi, 2021*).

Although AI has shown promising results in assisting clinicians in getting the best outcome, delivering healthcare services could be interrupted if the medical devices are not optimally operated. Critical medical devices are the main priority and critical areas for patients. Intensive care units (ICU) and hybrid COVID-19 wards are extensively used in treating patients with medical devices operated without failure. As a result, demand for critical care devices such as ventilators has skyrocketed (*Garzotto et al., 2020*; *Markets and Markets, 2021*). A smart healthcare system and efficient maintenance strategy can prevent potential failure or breakdown, disrupting healthcare operations and leading to serious patient injury. To date, the published works focused on developing a medical device reliability system specific to one type of device. Comprehensive maintenance and reliability system was recently published by *Hilmi et al. (2021)* and *Zamzam et al. (2021)*; however, the works focused on predicting maintenance prioritization. They highlighted a research gap of inconsistent mathematical methodologies requiring manual intervention in determining the weights of criteria in reliability assessments. It is necessary to improve the current predictive models for various medical devices. With adequate maintenance history in structured data to train a model in aiming for the best accuracy, the study on performance prediction for medical devices using AI for smart healthcare management can be further explored. Therefore, this article addresses three identified research gaps as follows:

i) To date, there are limited published works on smart medical device maintenance strategy frameworks. Only three research studies utilized AI in medical devices' performance prediction. The developed predictive models were proposed by analyzing only one device's history data: infant incubators, infusion pumps, and defibrillators. The comprehensive medical device reliability assessments are still lacking, and cost analysis was not considered in the assessment.

ii) To the best of our knowledge, medical device reliability studies were categorized into three main areas: risk assessment, performance or failure prediction, and management system. Most studies focus on risk management using failure codes analysis and maintenance prioritization in reliability assessments. Utilizations of smart management and monitoring leveraging extensive data capability are limited and have not been appropriately explored. A predictive model to forecast the likelihood of equipment failures is lacking. Anticipating these problems is essential in maximizing device uptime and reducing astronomical repair costs. This is crucial for any country during crisis management, especially during COVID-19 pandemic, where excellent and reliable equipment is highly required.

iii) Three research on performance prediction for medical devices is available using AI, as reported by *Badnjević et al. (2019)*, *Hrvat et al. (2020)* and *Spahić et al. (2020)*. The existing model predicts medical device performance by developing accurate and faulty classification based on the pass or fail response. Current practice is to perform

scheduled preventive maintenance, and the manufacturer suggests its frequency without considering failure history data. A scientific research gap is improved in this study with the development of a critical medical devices predictive model to predict the likelihood of device failure from its purchase date. The predictive model will be able to classify the device into three classes; (i) class 1, where the device is unlikely to fail within the first 3 years of purchase, (ii) class 2, where the device is likely to fail within 3 years from purchase date, and (iii) class 3, where the device is likely to fail more than 3 years after purchase. The machine learning and deep learning models were compared. The goal is to determine the actual maintenance schedule needs and gain comprehensive strategic maintenance management to reduce maintenance and operational cost.

# MATERIALS AND METHODS

## Predictive model framework

This study considers five types of healthcare facilities under the Ministry of Health, Malaysia. They are four types of hospitals: state, major, minor, non-specialist, and one special psychiatric institution, which is equivalent to 15 healthcare facilities. A predictive model was developed based on the medical device data (*Engineering Services Division Ministry of Health, 2018*) from government hospitals in Perak (west coast of Malaysia peninsula). Perak state hospital is equipped with 990 beds, two major specialist hospitals with 608 and 548 beds, two minor specialist hospitals with 305 and 250 beds, nine non-specialists or district hospitals with 50 to 160 beds, and 1,800 beds for a special psychiatric institution. The state hospital provides 15 specialist services and designated sub-specialists based on the region, and clinical service is managed by clustering in their respective areas. The 15 medical specialist services cover and are not limited to general medicine, general surgery, pediatrics, orthopedics, obstetrics and ophthalmology, ENT (otorhinolaryngology), emergency medicine, psychiatry, dental, dermatology, and nephrology. From these 15 facilities, there were 12,214 medical device units with active and inactive critical medical devices from 1997 to April 2021. The medical device maintenance at these 15 facilities is currently performed by a concession company appointed by the government in a long comprehensive contract. A challenge is encountered in gathering, integrating, maintaining, processing, and analyzing various types of medical data. It is too complicated and inefficient to handle using existing database management systems because it consists of big healthcare data. Although computerized maintenance management can store a vast number of data, difficulties occurred when clinical engineers could not provide myriad maintenance data into a robust tool that enables the implementation of comprehensive maintenance strategies. The utilized medical device database contains both structured and unstructured data. Structured data is the device's general information, whereas unstructured data includes routine maintenance service records and troubleshooting actions performed by competent personnel (*i.e.*, clinical

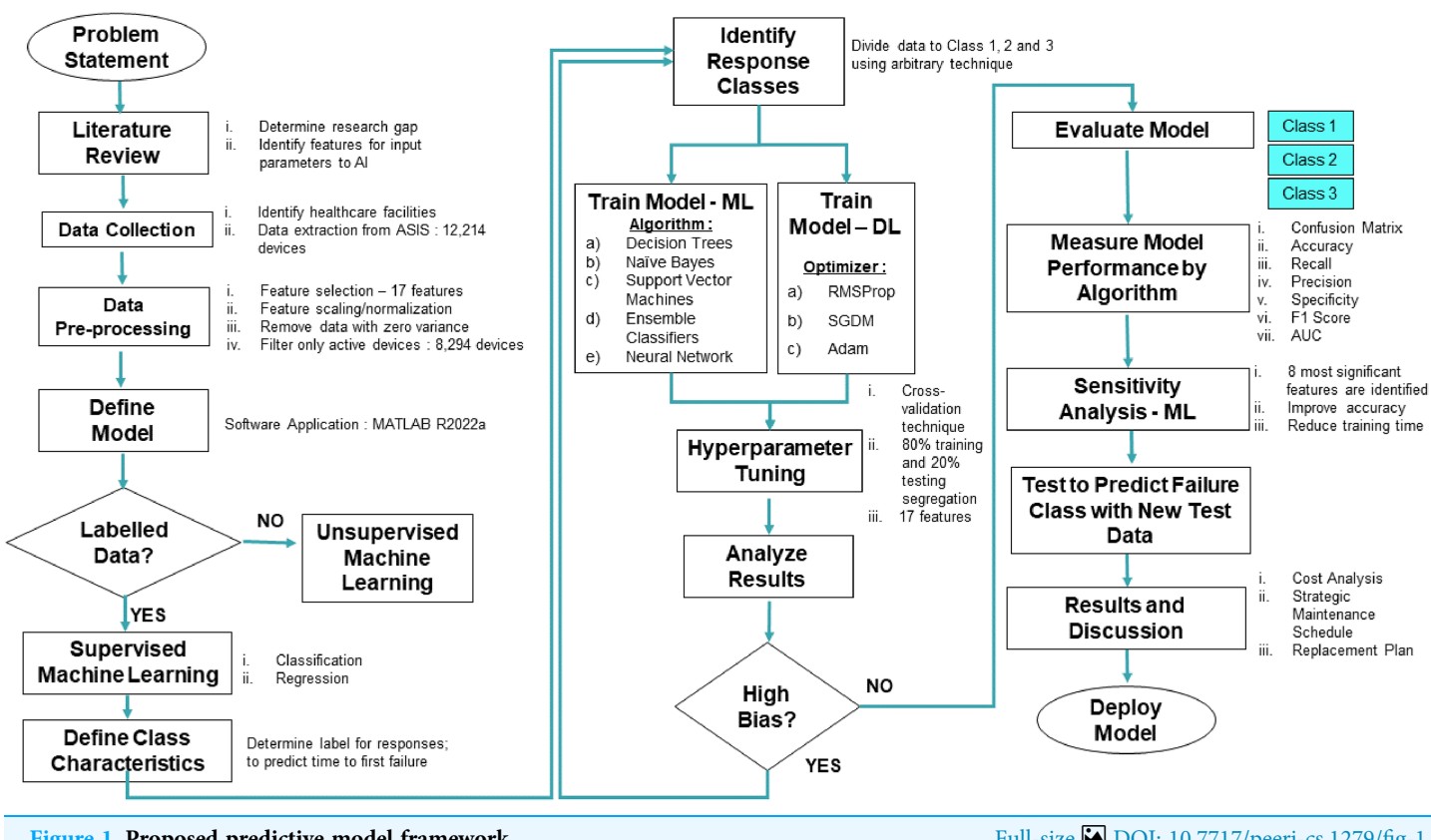

**Figure 1  Proposed predictive model framework.**                               

engineers). Utilizing unstructured data necessitates a laborious pre-processing phase to clean and organize the data.

## Predictive model development

Government hospitals in Malaysia used a web-based database to record and store medical device history since 1997. The web-based data for this analysis is asset and services information system (ASIS), which spans from the system's inception in 1997 to April 2021. Figure 1 depicts the overall proposed framework where data on 44 types of critical medical devices are extracted from 15 healthcare facilities (comprised of 8,294 devices). The pre-processing data stage is executed, where 17 input parameters are selected based on literature review findings. All 17 features from 8,294 devices are extracted to be fed to the proposed predictive model to anticipate the likelihood of first failure of the medical devices.

Normalization is required in the pre-processing stage, where this process will ensure all features are within the same scale and range. After normalization, data is combined with categorical data as an input parameter or features to the predictive model. Normalization returns the vector-wise *z-score* of the dataset with center zero (0) and a standard deviation of one (1). Normalization operates on each column of data separately, and the below equation is used for *z-score* of a value *x* (*Zamzam et al., 2021*);

$$z\text{-}score = \frac{x - \mu}{\sigma}$$

$$\sigma = \sqrt{\frac{\sum_{i=1}^{n}(x_i - \mu)^2}{n}}$$

($x$ = value, $\mu$ = mean, $\sigma$ = standard deviation, $n$ = highest probability estimate of the population's standard deviation)

The data point distance from the mean and standard deviation is the measurement for *z-score*. Returning as a vector and matrix, standard deviation of *x* and mean of *x* are used to calculate the *z-score*. After normalization is executed, a normalized value is imported into the software.

The following process flow is to identify the response classes boundary for classes 1, 2, and 3. The class is divided using an arbitrary technique based on the pattern in the data. The model is trained on five algorithms using the cross-validation technique with 80% training and 20% testing data segregation. Observing the confusion matrix, the model performance is examined, and the values for recall, precision, specificity, F1 score, and AUC are calculated. We perform sensitivity analysis to further optimize the developed predictive model in determining the most significant features for medical device failure prediction. The proposed model is then tested using a new test dataset to predict failure classes and the outcomes on cost impact. A new proposed maintenance schedule and replacement plan will be discussed further in the discussion section.

## Machine learning and deep learning application

In this stage, ML and DL techniques are explored using five algorithms and three optimizers for both approaches, respectively (Fig. 1). Both ML and DL techniques were compared with the 17 features embedded in the model. A sensitivity analysis is performed to optimize the reliability and rank the significant features in determining the best predictive model. Decision trees, naïve Bayes, support vector machines, ensemble classifiers, and neural network algorithms are used for ML applications. Meanwhile, the stochastic gradient descent with momentum (SGDM), root mean square propagation (RMSProp), and adaptive moment estimation (Adam) are applied for DL. A support vector machine in ML splits data into classes by locating the optimal hyperplane that divides each point into its corresponding category. Conversely, numerous weak learners' outputs are combined into one accurate ensemble model using ensemble classifiers algorithm by boosting the maximum number of splits and learners. A similar tree model is also applied in decision trees with responses predicted by following the root to leaf node and containing responses in true or false conditions. Gaussian distribution with a mean and standard deviation is used in naïve Bayes to simulate the predictor distribution within each class. In addition, a feedforward fully connected neural network is utilized, which has

a connected layer with each fully linked layer that multiplies the input by a bias vector and a weight matrix.

Moreover, a convolutional neural network (CNN) for DL is used with SGDM, RMSProp, and Adam as the training options. SGDM optimizer specifies the momentum value using momentum training options, and RMSProp has a decay rate option using squared gradient decay factor. Adam optimizer is propounded with decay rates of gradient besides squared gradient moving averages using gradient and squared gradient decay factor. As a result, the SGDM optimizer can oscillate along the precipitous descent, leading to the best result. One method to decrease this oscillation is to include a momentum term in the parameter update as specified in the SGDM equation below (*Dubey et al., 2020*; *Essai Ali & Taha, 2021*; *Setiawan et al., 2019*; *Sultana et al., 2021*):

$$\theta_{\ell+1} = \theta_\ell - \alpha \nabla E(\theta_\ell) + \gamma \left( \theta_\ell - \theta_\ell - 1 \right)$$

($\gamma$ = current iteration's contribution from previous gradient step, $\alpha$ = learning rate, $\ell$ = iteration number, $\theta$ = parameter vector, $\nabla E(\theta)$ = loss function).

The application of learning rates that vary by parameter and may automatically adjust to the optimized loss function shall enhance the network training. RMSProp comes into place where it upholds a moving average of the parameter gradients' element-wise squares with a decay rate of the moving average identified as β2 and is applied in the below equation (*Dubey et al., 2020*; *Essai Ali & Taha, 2021*; *Setiawan et al., 2019*; *Sultana et al., 2021*):

$$v_\ell = \beta_2 \, v_{\ell-1} + (1 - \beta_2)[\, \nabla E(\theta_\ell)\,]^2$$
$$\theta_{\ell+1} = \theta_\ell - \frac{\alpha \nabla E(\theta_\ell)}{\sqrt{v_\ell} + \in}$$

($\beta_2$ = the moving average's rate of decay, $\in$ = small constant is added to prevent zero division).

Besides, RMSProp has similar characteristics to Adam, provided Adam has the added momentum term. It maintains a moving average, element by element, of the parameter gradients and their squared values using the below equation (*Dubey et al., 2020*; *Essai Ali & Taha, 2021*; *Setiawan et al., 2019*; *Sultana et al., 2021*):

$$m_\ell = \beta_1 m_{\ell-1} + (1 - \beta_1) \nabla E(\theta_\ell)$$
$$v_\ell = \beta_2 \, v_{\ell-1} + (1 - \beta_2)[\nabla E(\theta_\ell)]^2$$
$$\theta_{\ell+1} = \theta_\ell - \frac{\alpha m_\ell}{\sqrt{v_\ell} + \in}$$

($\beta_1$ = gradient decay rate).

A decay rate value of β1 and β2 can be specified using the gradient decay factor. Adam optimizer uses a moving average, and network parameters are updated using the equation. When gradients over several iterations are comparable, employing a moving average of the gradient allows the parameters to change and gain momentum in a particular direction. The parameter updates also decrease in size if the gradient is primarily noise-based because

the moving average of the gradient shrinks. A technique to counteract a bias that occurs at the start of the training is also included in the whole Adam network.

## Features selection

Numerous critical medical devices are discussed in the Malaysia Standard of Good Engineering Maintenance Management of Active Medical Devices in MS2058: 2018 (*Department of Standards Malaysia, 2018*). They are categorized into three main groups: therapeutic, diagnostic, and laboratory. Besides, the devices are grouped into critical equipment and patient support equipment. This article selects the critical medical devices as listed in MS2058: 2018. From this list, 72.4% of therapeutic and 48% of diagnostic devices are categorized as critical devices, whereas these two categories also increased efficacy, complexity, and tendency for adverse effects (*Curtis, Tzannes & Rudge, 2011*). Despite its criticality, in 2020, the diagnostic imaging equipment segment held the greatest market share, significantly impacting the medical device maintenance market globally (*Markets and Markets, 2021*). All devices under the laboratory group are categorized as patient support equipment and not classified as critical medical devices. Hence, by eliminating inactive medical devices and laboratory medical devices, the total number of 8,294 active medical devices. All 44 types of critical devices are located at critical locations such as operation theatre (OT), ICU, accident and emergency (A & E), and wards. A total of 34.68% of the devices are infusion pumps, and 14.93% are physiologic monitoring systems which are located in various areas in Perak hospitals. Other devices below 10% in percentage are radiographic systems, drills bone, cystoscopes, colonoscopes, colposcopes, laparoscopes, mobile radiographic/fluoroscopic, dental radiographic, surgical hand drills, injectors, lithotripters, pacemakers, peritoneal dialysis units, resuscitators, stimulators, and ultrasonic. High-end medical devices or other categories are small in numbers and primarily located in X-ray Department. These include Radiographic/Fluoroscopic Systems Angiographic/Interventional, Radiographic/Fluoroscopic Systems General-Purpose Radiographic Units Mammographic, Scanning Systems Computed Tomography, Scanning Systems Magnetic Resonance Imaging, Full-Body. This high-end equipment is located at State Hospitals, Major Specialist Hospitals, and Minor Specialist Hospitals with medical imaging practitioners or Radiologist specialists.

We have selected 17 input parameters or features to be fed and tested in ML and DL predictive frameworks. Details on each parameter are summarised in Table 1, where some features are entered numerically, and others are provided to the ML and DL framework in a categorical manner.

### Service support

In manufacturing medical devices, a high technological impact is implemented. Therefore, manufacturers or authorised representatives must ensure the technical service's availability throughout its life cycle. After-sales service is essential for troubleshooting during failure, including procurement of spare parts, including wear and tear components, and the said parties shall provide a technical recommendation. During the procurement stage, the user and technical procurement committee will request the medical device suppliers to include

**Table 1 Input parameters for critical medical devices classification.**

| Predictor | Description | Values | Type |
|---|---|---|---|
| Hospital code | Hospital code identification in ASIS | 15 different codes (*e.g.*, PRK300, PRK301) | Categorical |
| Type description | Type of medical device | 44 different types (*e.g.*, Aspirators) | Categorical |
| Age | Current age of the device in years | Unique values | Numerical |
| Service support | The availability of service from manufacturer or authorized vendor | 1: End of production<br>0: Service available | Numerical |
| Asset condition | Current device condition | 0: Active/in use<br>1: Unrepairable failure but still in use<br>2: Approved for disposal | Numerical |
| Service intention | Device group | 1: Diagnostic<br>2. Therapeutic<br>3. Life Support | Numerical |
| Frequency maintenance requirement | PPM schedule as per manufacturer requirement | 1: PPM, yearly<br>2: PPM, twice-yearly<br>3: PPM with quality control certificate from class H licensee twice-yearly) | Numerical |
| Maintenance complexity | Complexity in performing maintenance procedures | 1. Average maintenance with EST<br>2. High-end maintenance with EST | Numerical |
| Total downtime | Total downtime in hours for unscheduled maintenance inclusive of corrective maintenance, breakdown repair, and breakdown during warranty period | Unique values | Numerical |
| Alternative & backup | Alternative service or device replacement during failure. Loaner is provided from rental service or respective vendor | 0: No loaner provided<br>1: Loaner provided | Numerical |
| Operations | The average usage in hours as in MS2058. Based on average use, actual use requires usage log or sensor monitoring | 1: 12 h/6 days<br>2: 24 h/7 days | Numerical |
| Total maintenance cost | Total cost for unscheduled maintenance inclusive of corrective maintenance and breakdown repair. Only include expenses entered into the system, and the expense of sending a patient to a private hospital is not included. | Unique values | Numerical |
| Purchase date | The date of purchase | Unique values | Numerical |
| Make | The device's country of origin | 37 different countries (*e.g.*, Malaysia) | Categorical |
| Model | Device model | 1,375 different models (*e.g.*, Smartvent 7900) | Categorical |
| Manufacturer | Device manufacturer | 511 different manufacturers (*e.g.*, Datex-Ohmeda Inc) | Categorical |
| Brand | Device brand | 568 different brands (*e.g.*, Smartvent) | Categorical |

**Note:**

PPM, planned preventive maintenance; EST, electrical safety test.

a guarantee letter of product service for at least 10 years after delivery. Similarly, the manufacturer will publish a discontinuation letter to inform customers that the model has been discontinued after 10 years or less. At this point, the user is urged to replace or upgrade the medical devices.

### Asset condition

Asset condition definition is divided into three categories: active or still in use, declared as an unrepairable failure, and approved for disposal. The same terminology is used in ASIS to categorize these devices. A functioning device being utilized by patients or placed on standby in clinical locations is an active device. Whenever after-sales service is no longer available and/or securing spare parts is problematic, the Concessionaire's maintenance team will recommend the device be classified as an unrepairable failure. On this note, clinical engineers will evaluate the recommendation to verify whether there is a shortage of medical devices, the device's safety, and replacement is approved upon disposal. Later, the clinical engineer will issue a disposal certificate to proceed with the disposal process as per the Malaysia Treasury Circular.

### Service intention of function

Service intention refers to the medical equipment's primary purpose or intended use. Five criteria are involved: life support, therapeutic, diagnostic, analytical, and miscellaneous (*Zamzam et al., 2021*). A life support device is the type of device of which patients would suffer prolonged or new injury, or worse still, be in a fatal state, should the device become unavailable or malfunction. The device that provides treatment for any illness or disease is therapeutic equipment. A diagnostic device is used for diagnostic purposes, and the device will display clinical parameters or human anatomy images for further diagnosis that will be examined by clinical personnel. This study excludes the analytical and miscellaneous categories since only critical groups are included.

### Frequency maintenance requirement

The manufacturer of the device sets frequency maintenance requirements for every device. As they become old or highly utilized, more maintenance is required, and subsequently high risk of maintenance errors (*Dhillon & Liu, 2006*). Upon procuring a medical device, the manufacturer will provide a user and service manual with a suggested maintenance schedule. The suggestion offers PPM frequency, spare parts code, minor troubleshooting steps, *etc.* In Malaysian Government hospitals, the Concessionaires will follow the maintenance schedule set by the manufacturer, which is executed in a cycle of three months, semi-yearly or yearly. Besides, the medical device regulation in Europe establishes a new reusable or reprocessed class I device where the manufacturer must support the safety and efficacy of the cleaning, disinfection, and sterilizing processes (*Garzotto et al., 2020*). Diagnostic imaging devices shall fulfill the requirement set by the regulation due to harmful radiation effects and must be controlled within a specific limit (*Anis et al., 2020*; *Atomic Energy Licensing Board, 2006*). The quality control (QC) certificate issued by a class H Licensee must be issued twice a year, generally after PPM is conducted, to ensure that radiation exposure is within safe limits for patients. Maintenance at the specified intervals can reduce the likelihood of failure, but it requires a high cost and skilled employees trained by the manufacturer.

### Maintenance complexity

The degree of difficulty in completing maintenance procedures is defined as maintenance complexity. There are two types of maintenance complexity set as predictors: average and high-end maintenance with electrical safety test (EST). High-end equipment is principally diagnostic imaging equipment and system located in radiology department. This type of system requires a three-phase power supply and is equipped with a console and voltage stabilizer as a complete system. EST is performed on a device with direct contact with patients or humans to ensure no electrical leakage, and the current flowing is within the limit. Electrical safety is paramount in medical device quality assurance best practices. Shock can cause disruptions during healthcare procedures and result in injury or death. The main objective of this test is to ensure patient and user safety.

### Total downtime

Downtime is a reverse condition of uptime. A medical device in a non-functioning state is under a downtime period until the rectification work is completed. The downtime affects the service delivery, and the user needs to search for a replacement or standby unit during the interrupted period. The time will be calculated from the time user launch a complaint through the helpdesk or ASIS, where the time is recorded until the device is back to its normal condition. In Malaysia's medical device service contract, the Concessionaire shall provide a replacement or loaner during this downtime period. To ensure service delivery is in place, they must also bear the cost of outsourcing patients or laboratory samples to private hospitals as a service delivery obligation.

### Alternative and backup

Alternative services are a mechanism to minimize service interruption where an external party or private healthcare provider provides patient care services. This includes outsourcing patients or laboratory samples due to faulty medical devices and maintenance factors. A loaner or a backup device is a medical device temporarily placed on service to replace the malfunctioning device to ensure minimal service interruption. The replacement must be executed after failure and limited to the critical medical device set in the contract. A supply of loaners is a must in the comprehensive contract for selected medical devices such as aspirators, bronchoscopes, colonoscopes, cardiotocographs, defibrillators, electrocardiographs, hemodialysis units, incubators, infusion pumps, mattress systems, nebulizer, vital sign monitors, physiologic monitoring systems (acute care), electrosurgical unit, sphygmomanometers, and ventilators. Other devices are not included in the contract, and a replacement request is an option to be fulfilled.

### Operations

The average usage in hours within 6 and 7 days is defined as a parameter for operations features. There are two types of operations: average usage of 12 h in 6 days or 24 h in 7 days. The time is an average based on locations and is subject to the actual usage by the user. Unfortunately, no sensor has been installed to monitor every medical device's usage or utilization. The more accurate result for this feature requires the installation of an individual sensor at every device for recording purposes. A sensor installation and usage

log are still under the Ministry of Health Malaysia's ongoing projects, and the model shall be improved after the installation is completed.

### Maintenance cost

Maintenance cost is the total cost spent by the Concessionaire to rectify the failure of medical devices. The cost will be entered into the system for record purposes and spare part tracking or analysis. Providing all maintenance services in-house is not always possible. In such cases, using external service providers for a significant portion of the maintenance activities may be necessary; thus, this will incur additional costs. External service providers are divided into two categories which are equipment manufacturers and independent service organizations (*World Health Organization, 2011b*). The maintenance cost in this article includes contractor costs if authorized vendors, labour, and parts costs perform the rectification work. The data will consist of only the cost available in the system, and the cost of outsourcing patients or samples is not included.

### Manufacturing country, model, manufacturer, and brand

Despite device utilization and age, other essential factors in evaluating a medical device's performance are model, manufacturer, manufacturing country, and brand. For example, two medical devices of the same age will have different performance and uptime status, depending on the model specification and utilization. These features help train a model to evaluate performance based on its specification, and the proposed model can be used by tendering committee during the medical device procurement stage in the future. The difference between these four features is that the USA is the country of origin, and the model is Smartvent 7900. The manufacturer is Datex-Ohmeda Inc, and the Smartvent represents the Brand. The major players in the medical devices maintenance market are GE Healthcare (Chicago, IL, USA), Siemens Healthineers (Erlangen, Germany), Koninklijke Philips N.V. (Amsterdam, Netherlands), Medtronic (Dublin, Ireland), and Fujifilm Holdings Corporation (Minato City, Japan), Canon, Inc. (Toshiba Medical System Corporation, Otawara, Japan), Agfa-Gevaert Group (Mortsel, Belgium), Carestream Health, Inc. (Rochester, NY, USA), Drägerwerk AG & Co. KGaA (Lübeck, Germany), Hitachi Medical Corporation (Tokyo, Japan), Althea Group (Milano, Italy), Olympus Corporation (Shinjuku City, Japan), B. Braun Melsungen AG (Melsungen, Germany), KARL STORZ GmbH & Co. KG (Tuttlingen, Germany), and Aramark Services, Inc. (Philadelphia, PA, USA) (*Markets and Markets, 2021*).

## RESULTS

### Failure classes

The classification problem in supervised ML is overcome by defining response classes before execution. Hence, based on the tabulated data extracted from ASIS, an arbitrary technique is used to create three classes based on the available dataset. Then, the best balance classes are selected, and the response classes for critical medical devices are subdivided into classes 1, 2, and 3. With the capability of supervised machine learning analyzing retrospective data, classifying critical medical devices is beneficial for more effective PPM planning, management replacement plan, and strategic annual budget

**Table 2 Predicted failure classes based on few criteria after classification.**

| Classes description | Class 1 | Class 2 | Class 3 |
|---|---|---|---|
| Class description | 0 failure | ≤1–36 months to first failure | ≥37 to 1,440 months to first failure |
| Type description | 36 types of critical medical devices | 45 types of critical medical devices | 37 types of critical medical devices |
| Total number of devices | 2,231 devices | 4,107 devices | 1,956 devices |
| Age | <1 year–30 years | 1–27 years | 3–30 years |
| Service support | Service available and end of production | Service available and end of production | Service available and end of production |
| Asset condition | Active, unrepairable failure and approved for disposal | Active, unrepairable failure and approved for disposal | Active, unrepairable failure and approved for disposal |
| Service intention | Diagnostic, therapeutic and life support | Diagnostic, therapeutic and life support | Diagnostic, therapeutic and life support |
| Frequency maintenance requirement | PPM yearly, twice yearly, and with quality control certificate | PPM yearly, twice yearly, and with quality control certificate | PPM yearly, twice yearly, and with quality control certificate |
| Maintenance complexity | Average maintenance with EST only | Average maintenance with EST and high-end maintenance with EST | Average maintenance with EST and high-end maintenance with EST |
| Total downtime | 0.00 | Up to 25,743 h | Up to 10,852 h |
| Alternative and backup | No loaner | No loaner and loaner provided | No loaner and loaner provided |
| Operations | 12 h/6 days and 24 h/7 days | 12 h/6 days and 24 h/7 days | 12 h/6 days and 24 h/7 days |
| Total maintenance cost | MYR 0.00 | 0 to MYR 1,368,803.83/device | 0 to MYR 123,674.03/device |
| Purchase date | 1991 to February 2021 | 1994 to 2020 | 1990 to 2017 |
| First failure date | No failure | 1997 to April 2021 | 1997 to April 2021 |
| Make/manufacturer country, model, manufacturer and brand | Varies | Varies | Varies |

preparation. As for newly purchased critical medical devices, the proposed model can evaluate its performance throughout its lifespan based on the brand, model, failure history, and other features used during the training stage. Five algorithms are utilized in ML: decision trees, naïve Bayes, support vector machines, ensemble classifiers, and neural networks. In addition, a cross-validation technique with segregation of 80% training and 20% testing is used.

The proposed predictive model uses 17 features, as tabulated in Table 1, to distinguish the devices into three classes. Class 1 is defined as unlikely to fail within the first 3 years from the purchase date, while class 2 is for devices that are likely to fail within 3 years. Class 3 is for devices likely to fail more than 3 years after purchase, as tabulated in Table 2. Medical devices in class 1 are less critical and have low complexity in maintenance than in classes 2 and 3. Class 1 can be identified as the least problematic devices, including newly purchased ones. Class 2 is a matter of concern since the failures are detected closer to the purchase date. The class 3 devices are classified as prone to failure after 36 months of purchase date. The scheduled maintenance frequency is conducted as per the manufacturer's recommendation in regular maintenance practice. There is no

**Table 3 Performance evaluation by algorithms using 17 features.**

| Technique | Algorithm | Accuracy | Recall | Precision | Specificity | F1 score |
|---|---|---|---|---|---|---|
| Machine learning | Decision Trees | 76.30% | 73.20% | 73.57% | 86.79% | 73.38% |
| | Naïve Bayes | 59.80% | 62.69% | 60.38% | 80.87% | 61.52% |
| | Support vector machines | 65.90% | 64.51% | 62.92% | 82.45% | 63.70% |
| | Ensemble classifiers | 77.90% | 74.67% | 75.39% | 87.60% | 75.03% |
| | Neural network | 69.90% | 66.64% | 68.12% | 83.35% | 67.37% |
| Deep learning | RMSProp | 68.03% | 64.70% | 65.04% | 82.29% | 64.87% |
| | SGDM | 70.33% | 67.11% | 67.15% | 83.71% | 67.13% |
| | Adam | 68.76% | 65.18% | 66.18% | 82.77% | 65.67% |

consideration has been made based on the failure history data. Therefore, with the proposed predictive framework in this study, new recommendations shall be made where maintenance frequency and cost can be reduced based on the forecasted analysis attained from the proposed model. Out of the total 8,294 critical devices in 15 different hospital categories, 49.51% of devices were classified as class 2, while the remaining 26.89% and 23.58% are categorized as Class 1 and 3, respectively. Thus, the yearly budget allocation for class 1 can be reduced and should be reallocated to classes 2 and 3.

## Parameter tuning and optimization

Parameter tuning and optimization are executed to improve classification performance, accuracy and introduce a unique identity to the model. Ensemble classifier outperforms other algorithms with an accuracy of 77.90%, followed by decision trees with 76.30% after parameter optimization, as explained in Table 3. Receiver operating characteristics (ROC) curves and AUC were measured where the higher the AUC values (as it is closer to '1') indicate an excellent predictive model (*Shiferaw, Bewket & Eckert, 2019*). In this study, Ensemble classifiers attained an AUC of 0.89, decision trees 0.85, while both support vector machines and neural networks attained 0.82. Meanwhile, naïve Bayes only achieves 0.80 for AUC values. Parameter optimization increased the model accuracy for four algorithms. However, parameter optimization for decision trees remains at an accuracy of 76.30% before and after optimization. Hence, ensemble classifier performs best with the highest recall, precision, specificity, and F1 Score values after parameter tuning and optimization. As compared to the DL technique, SGDM optimizer denotes the highest accuracy compared to RMSProp and Adam optimizer. There is a reduction in performance accuracy from 77.90% for ML to 70.33% for the DL model. However, DL has the advantage of shorter training time than ML. DL requires 1 min 5 s to complete the training progress, much faster than 11.49 min using ML.

A model optimization is performed for all algorithms and optimizers in both techniques. A kernel scale is optimized in support vector machines algorithm from automatic to one, producing 65.90% instead of 65.80% during pre-optimization, as in
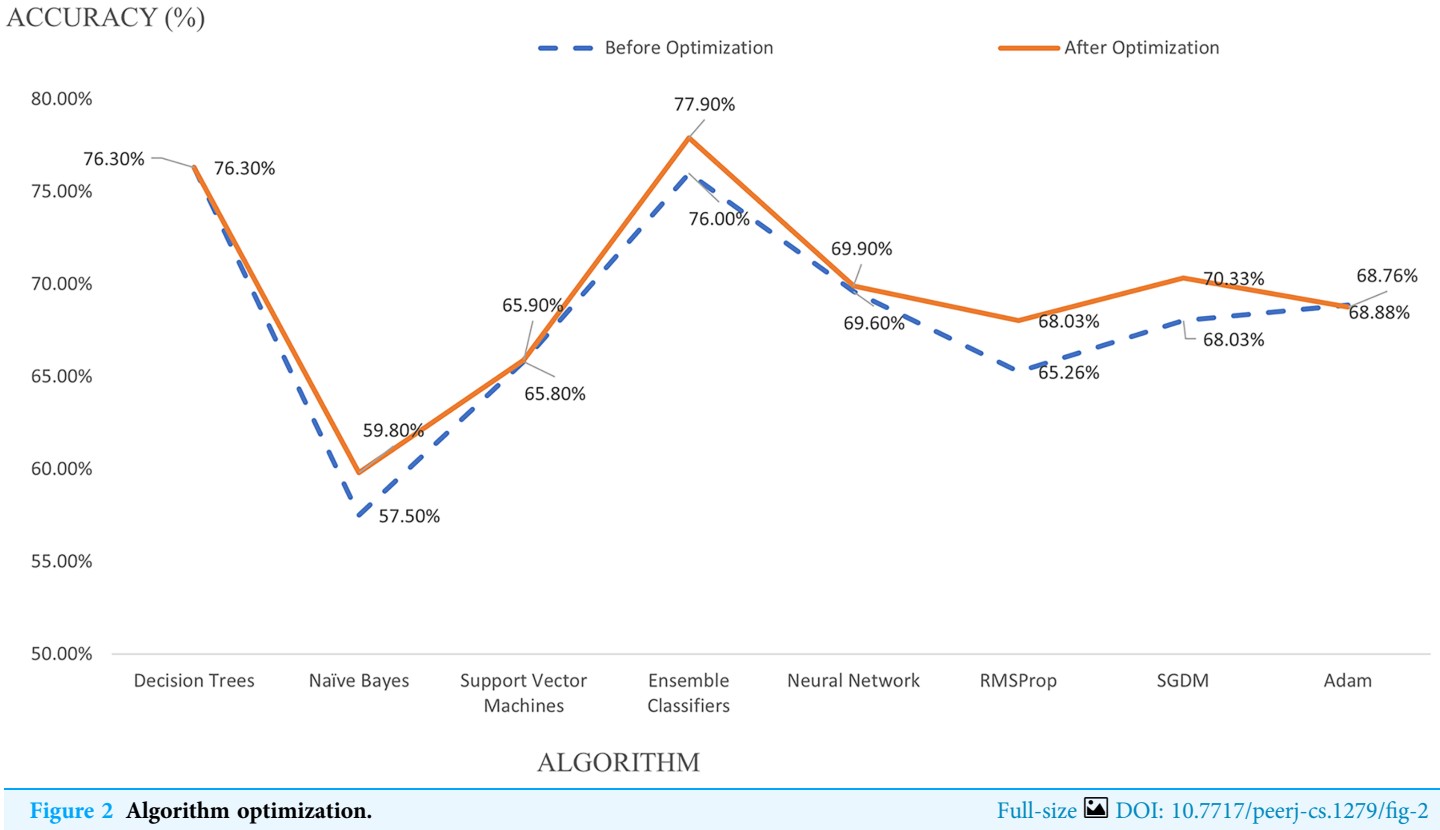

**Figure 2  Algorithm optimization.**                               

Fig. 2. The decision tree algorithm's accuracy remains at 76.30% when a maximum number of splits is tuned from 100 to 1. A different number of splits gives zero impact to the model but demands a higher training time from 4.1609 to 33.053 s. As for naïve Bayes, the accuracy increased from 57.50% to 59.80% by changing the categorical predictors to Gaussian parameters with a training time increase from 40.397 to 366.74 s. Meanwhile, the ensemble classifier improves its accuracy from 76.00% to 77.90% during the optimization stage when the maximum number of splits is reduced from 6,635 to 20, and ensemble method is set from Bag to Ada Boost. The number of learners is retained at 30 after optimization, and learning rate is set to 0.1 with an increasing training time from 16.575 to 689.92 s. Besides, a number of fully connected layers is optimized from 1 to 3, and first layer size is reduced from 100 to 10 for the neural network algorithm, improving the accuracy from 69.60% to 69.90%. However, neural network algorithm in supervised ML denotes the longest training time of 57,685 s.

The model optimization is applied for DL, where a tuning of mini-batch size from 128 to 100 with an increase of epoch from 30 to 60 gives a different performance impact to the model. SGDM optimizer uses a mini-batch size of 100, and a maximum epoch of 60 attained an accuracy of 70.33%, slightly lower than the ensemble classifier. The diminution of mini-batch size from 128 to 100 increases the model accuracy from 65.26% to 68.03% for RMSProp and 68.03% to 70.33% for SGDM optimizer, as in Fig. 2. Nevertheless, the

optimization process reduced the accuracy from 68.88% to 68.76% for Adam optimizer. A training progress graph interprets a DL technique, representing the accuracy of each unique mini-batch. The model performance can be monitored in real-time and model progress can be stopped at any time. Figure 3 shows RMSProp takes the longest elapsed time of 1 min 13 s, followed by 1 min 12 s for Adam and 1 min 5 s for SGDM optimizer, respectively. The model accuracy is plotted in "blue" while model losses are plotted in "orange." Initially, when the epoch increases, the losses are reduced and continually saturated until it reaches the maximum epoch of 60. The training progress is stopped at epoch of 60 since the progress is saturated and gives an insignificant impact if the process continues.

## Predictive model performance and evaluation

The quality of the characteristics given to the algorithm determines the accuracy of the model's predictions. The overall accuracy and Kappa coefficient are two extensively used indicators for thematic accuracy controls on error matrix (*Garcia-Balboa et al., 2018*). In addition, precision and recall are two relevant metrics used for evaluating prediction accuracy (*Teo et al., 2020*). The recall performance measures the trustworthiness of the result or ability to classify positive outcomes at a true positive rate. At the same time, the precision demonstrates the predictive or positive values which correctly predicted (*Kareen, 2020*). Both precision and recall are frequently at odds, and boosting precision reduces the recall with text retrieval concepts representing both calculations (*Ghorbani et al., 2020*). A recall is a number of relevant features in the selected subset divided by the total number of relevant features. For precision, it is divided by the total number of features specified in the dataset (*Santra & Christy, 2012*).

Analyzing and comparing models based on recall and precision is time-consuming; thus, employing the F1 score method is another viable result (*Ghorbani et al., 2020*). The F1 score is a harmonic average of precision and recall, consider both metrics and evaluate the model's accuracy and dependability. The confusion matrix is the most common method of reporting on the thematic accuracy of geographic data (*Garcia-Balboa et al., 2018*; *Santra & Christy, 2012*). The true class is represented by the rows of the confusion matrix, while the columns represent the predicted class in the 3 × 3 matrix. Correctly classified observations are expressed by diagonal cells; in contrast, erroneously classed observations are represented by off-diagonal cells with TN as a true positive, FP as a false positive, FN as a false negative, and FP as a false positive (*Kareen, 2020*; *Zamzam et al., 2021*). An equation or an evaluation metric by *Hameed et al. (2021)* is used to calculate accuracy, recall, precision, specificity, and F1 score based on a confusion matrix (*Kareen, 2020*; *Teo et al., 2020*). The below equation is calculated based on confusion matrix values and is summarized in Table 3. Ensemble classifier has the best performance with 74.67%, 75.39%, 87.60%, and 75.03% for recall, precision, specificity, and F1 score values, as explained in Table 3 previously.

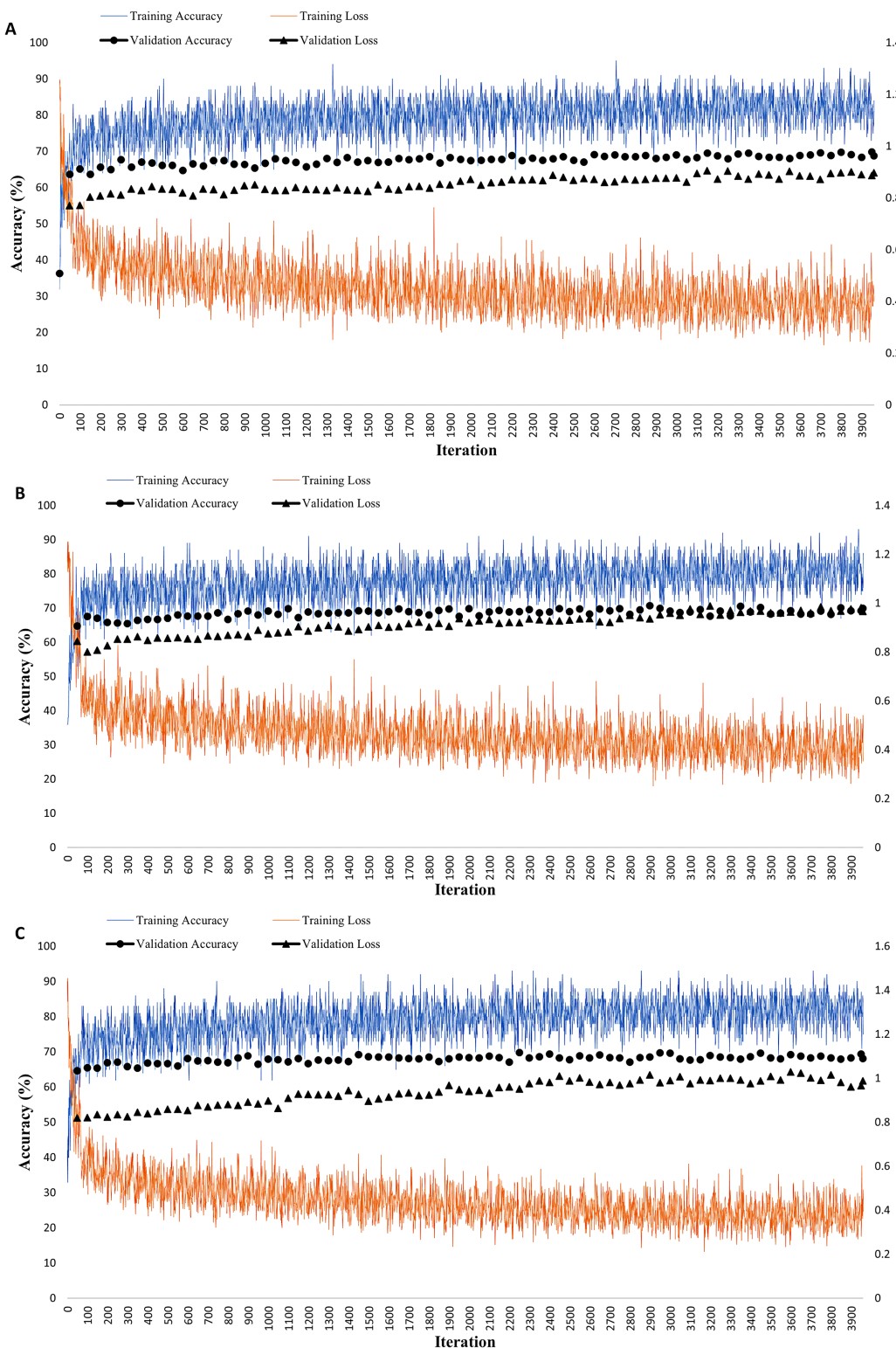

**Figure 3 Deep learning techniques with three different optimizers (A) RMSProp (B) SGDM (C) Adam.** The graph represents the accuracy (%) and loss, where the blue graph is the accuracy for network training, and the orange colour is the network loss. The black colour graph is the validation graph with the final epoch reaching 60 cycles.

$$Accuracy = \frac{Total\ number\ of\ correctly\ classified}{Total\ number\ of\ observation} = \frac{TP + TN}{TP + FP + TN + FN}$$

$$Sensitivity/Recall = \frac{TP}{TP + FN}$$

$$Precision = \frac{TP}{TP + FP}$$

$$Specificity = \frac{TN}{TN + FP}$$

$$F1\ Score = 2 \times \frac{Precision \times Recall}{Precision + Recall}$$

## Sensitivity analysis

Table 4 describes the sensitivity analysis and evaluates the features' effect on the model performance. Leave one out technique is performed by calculating misclassification and then dividing with all feature errors to obtain the ranking ratio (*Chen et al., 2020*; *Gazzaz et al., 2012*; *Pastor-Bárcenas et al., 2005*). The ratio is ranked in descending order to highlight the impact of the features on the model. Next, five sensitivity analysis techniques, namely Leave One Out, MRMR, Chi2, ANOVA, and Kruskal Wallis, are applied to compare the ranking for all features. MRMR utilizes the minimum redundancy maximum relevance algorithm to rank the characteristics in order. A chi-squared algorithm uses a chi-square test, an ANOVA uses one-way variance analysis, and Kruskal Wallis performs a hypothesis with the same median from the population (*The MathWorks Inc, 1994–2021*, *2022*). These three algorithms ranked the features by $-\log(p)$ scores, where the $p$-values denote the chi-square test statistics. The five highest-ranking features for every technique are underlined in Table 4, and the features are analyzed in the software to achieve better accuracy. Of the 17 features, only eight are identified as the most significant to obtain the highest accuracy. If more than eight features are selected, the accuracy decreases.

Figure 4 compares sensitivity analysis techniques when the eight most significant features are selected to develop the model. The features are age, service support, asset condition, maintenance complexity, total downtime, maintenance cost, model, and purchase date. The graph demonstrates ANOVA and MRMR outperform other techniques with 79.20% accuracy, followed by 79.00% for Chi2. Kruskal Wallis and the Leave one out technique attained 78.10% and 77.50% accuracy, respectively. The model is improved from 77.90% using 17 features to 79.20% accuracy with eight features after sensitivity analysis.

## DISCUSSION

### Proposed predictive model using machine learning

The ML predictive model is improved from 77.90% to 79.20% accuracy after sensitivity analysis, as described in the result section. The ML requires more training time of 11.49 min compared to only 1 min 5 s using DL when all 17 features are embedded in the model development. After eight significant features are imported with a learning rate of 0.01 and iterations of 50, the ML model improves its performance to 79.50% accuracy,

**Table 4 Five different techniques for sensitivity analysis.**

| Ensemble classifier | Leave one out ratio | Leave one out rank | MRMR score | MRMR rank | Chi2 score | Chi2 rank | ANOVA score | ANOVA rank | Kruskal Wallis score | Kruskal Wallis rank |
|---|---|---|---|---|---|---|---|---|---|---|
| Downtime | 1.357 | 1 | 0.381 | 2 | Inf | 2 | 187.2700 | 3 | Inf | 2 |
| Asset condition | 1.037 | 2 | 0.099 | 5 | 95.5770 | 12 | 86.9379 | 7 | 95.6105 | 8 |
| Maintenance complexity | 1.019 | 3 | 0.147 | 3 | 6.3473 | 16 | 6.3505 | 16 | 6.3463 | 16 |
| Operations | 1.015 | 4 | 0.002 | 17 | 2.0053 | 17 | 2.0050 | 17 | 2.0050 | 17 |
| Age | 1.014 | 5 | 0.082 | 6 | Inf | 1 | Inf | 2 | Inf | 3 |
| Make | 1.069 | 6 | 0.033 | 13 | 289.989 | 9 | 124.5004 | 5 | 121.672 | 6 |
| Hospital code | 1.009 | 7 | 0.018 | 14 | 44.9490 | 13 | 21.5687 | 13 | 24.9444 | 14 |
| Alternative backup | 1.007 | 8 | 0.014 | 16 | 37.3888 | 14 | 37.5841 | 10 | 37.3832 | 10 |
| Brand | 1.001 | 9 | 0.056 | 10 | 497.457 | 6 | 36.0254 | 11 | 35.0889 | 12 |
| Purchase date | 0.982 | 10 | 0.075 | 7 | Inf | 3 | Inf | 1 | Inf | 1 |
| Type description | 0.978 | 11 | 0.037 | 12 | 353.127 | 8 | 90.1450 | 6 | 105.273 | 7 |
| Manufacturer | 0.977 | 12 | 0.071 | 8 | 621.334 | 5 | 6.3681 | 15 | 12.2941 | 15 |
| Service intention | 0.972 | 13 | 0.048 | 11 | 180.234 | 10 | 43.1858 | 8 | 61.6585 | 9 |
| Maintenance cost | 0.972 | 14 | 0.116 | 4 | 495.091 | 7 | 20.8254 | 14 | 388.007 | 4 |
| Service support | 0.966 | 15 | 0.066 | 9 | 131.147 | 11 | 133.7483 | 4 | 131.127 | 5 |
| Model | 0.962 | 16 | 0.487 | 1 | 658.457 | 4 | 42.1680 | 9 | 37.3716 | 11 |
| Frequency maintenance requirement | 0.957 | 17 | 0.018 | 15 | 32.1296 | 15 | 35.4916 | 12 | 32.8471 | 13 |

**Note:**
The underlined number indicates the five highest-ranking features for every technique.

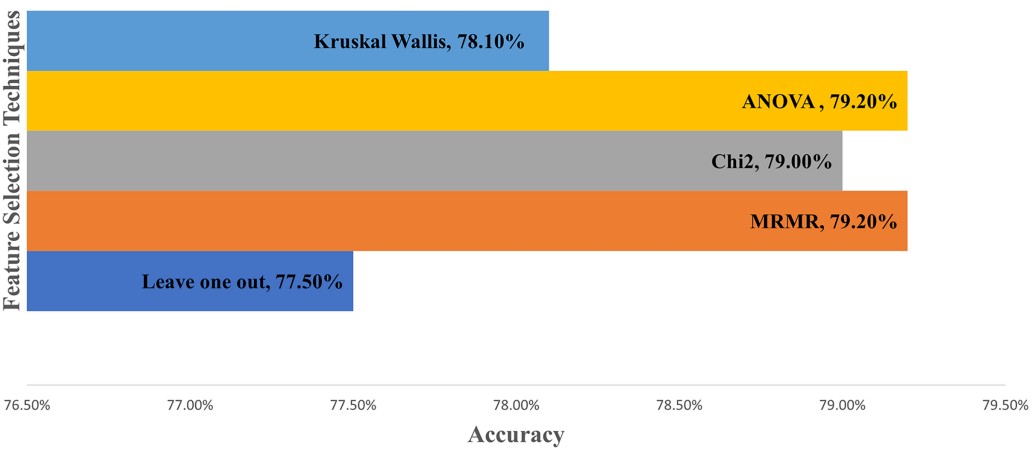

**Figure 4 Comparison between feature selection techniques in machine learning.**

76.05% recall, 77.43% precision, 88.36% specificity, and 76.73% for F1 Score indicator. The tuning of learning rate and iterations enhanced the model, with training time reduced from 11.49 to 7.908 min after sensitivity analysis. Although ML has better accuracy than DL, it

| Table 5 Parameter setting for ensemble classifier. | | |
|---|---|---|
| **Technique** | **Algorithm** | **Parameter setting** |
| Machine learning | Ensemble classifier | Accuracy: 79.50% |
| | | Specificity: 88.36% |
| | | Feature selection: MRMR |
| | | Ensemble method: AdaBoost |
| | | Maximum number of splits: 20 |
| | | Number of learners: 30 |
| | | Learning rate: 0.01 |
| | | Prediction speed: ~450 obs/s |
| | | Training time: 7.908 m |

requires more training time. The DL technique predicts more extensive data better, is primarily used in time series or image data and requires less time than ML. The result concludes that ML performs better in accuracy and all other performance indicators than DL. This best-optimized ensemble classifier model uses Ada Boost with a maximum number of splits of 20 and a learning rate of 0.01 to yield 79.50% accuracy, as shown in Table 5. The proposed model is expected to improve the current system toward smart healthcare management.

The model predicts classes 1, 2, and 3, which represent the time to the first failure event for an effective maintenance schedule and to utilize budget allocation. As compared to the other related works on medical device performance prediction, *Kovačević et al. (2020)* in infant incubators study predicted the device functionality and classified two different classes: accurate and faulty class with an accuracy of 98.5%. A similar methodology is applied by *Badnjevic et al. (2017)* for a mechanical ventilator. Using performance parameter values, a defibrillator study achieved 100% accuracy in the Random Forest classifier to predict positive: for devices that passed inspection or negative: for faulty devices (*Badnjević et al., 2019*). Meanwhile, *Hrvat et al. (2020)* attained 98.06% accuracy based on a conformity assessment where the outcomes are identified as pass or fail for infusion and syringe pumps. This article has significantly contributed to the medical device reliability assessment research by including 44 types of critical medical devices during model development and analyzing significant cost reduction after implementing the predictive model. To the best of our knowledge, no previous studies proposed a predictive model for anticipating the likelihood of a device's first failure after the purchase with cost analysis and comparison between ML and DL techniques.

## Characteristics for classes, schedule maintenance, and replacement plan

Classes 1, 2, and 3 have different properties, and the boundaries are set based on the patterns from the vast data. The present service contract fee per month in Malaysia is calculated by multiplying the device purchase cost and rate of fee, with the rate of fee defined in percentage based on the type of device. High-end equipment has a higher fee

rate equivalent to 19.25%, and the lowest rate is 4.95% per year for small devices (*Engineering Services Division Ministry of Health, 2018*). The fee includes PPM and corrective maintenance (CM) performed by Concessionaire. It is calculated in lump sum fees regardless of the number of failures events the devices encountered throughout their lifespan. A flat rate is imposed from the purchase date to the end of the device life. As a result, the Concessionaire gains a higher profit during the early age of devices, and the profit margin is reduced and approaches breakeven as the age rises.

$$Service\ contract\ fee/month\ = \frac{Purchase\ cost\ (MYR)\ \times Rate\ of\ fee\ (\%)}{12}$$

Poor maintenance, planning, and management are the most common causes of medical device failures (*Arab-Zozani et al., 2021*). Countries have different approaches to organizing maintenance and operational budget planning for medical devices. Saudi Arabia uses life cycle cost estimation in making decisions (*World Health Organization, 2011a*), and Turkey divided medical devices into technological groups to calculate cost distinctly (*Bektemur et al., 2018*). In the UK, they use purchasing, donations, replacement, and disposal policies to decide where, what, and when to procure medical devices. The replacement budget per year is implied by dividing the device's current prices by lifetime (*Temple-Bird et al., 2005*). The New Delhi Medical Equipment Maintenance policy uses the maintenance cost index as an indicator by dividing maintenance cost by capital cost. The maintenance cost values should not increase by 80% of the capital cost of equipment (*Kumar, 2012*). The United States of America applies the cost of service ratio by dividing the total annual cost for maintenance by the initial cost value. It is used as guidance to improve performance (*World Health Organization, 2011b*). Overall, at the time of speaking, none of these countries utilize AI applications for maintenance budget planning.

A rule of thumb with 80/20 rule is used for CM and PPM in *Stenström et al. (2015)*, and their results describe PPM represents 10% to 30% of total budget allocation compared to CM. This article's cost-saving calculation uses 80/20 for CM/PPM yearly and 70/30 for CM/PPM bi-annually for cost estimation analysis. class 1 age ranges between less than 1 year to 30 years, which consists of 2,231 devices in service. There is zero failure, and zero parts cost recorded throughout its lifespan. Throughout the years, PPM is scheduled annually and bi-annually for class 1, with a percentage number of devices are 90.45% and 9.55%, respectively. Therefore, due to its low criticality, PPM frequency is suggested to be reduced from bi-annually to annually with an estimation of cost-saving equal to MYR 199,256.45 per year, as demonstrated in Table 6. On this note, this study proposes a new maintenance strategy, which includes recommendations as illustrated in Table 7. Among others, this study suggests that by reducing the fee rate for class 1 devices, the operational cost can be further reduced. The PPM can be replaced with routine inspection for small devices such as aspirators where the maintenance tasks are minimal; hence the yearly maintenance budget allocation for this group can be reduced for the following year. Maintenance can be executed in-house where only average maintenance complexity with

**Table 6 Cost comparison for present and proposed service contract.**

| Class | No. of devices | Recommendation of this study | Present service contract cost MYR/year | Proposed service contract cost MYR/year | Cost saving MYR/year |
|---|---|---|---|---|---|
| 1 | 2,231 | Revise PPM frequency from bi-annually to annually | 2,838,246.45 | 2,638,990.00 | 199,256.45 |
| 3 | 1,956 | Revise PPM frequency from bi-annually to annually | 2,440,926.61 | 2,313,852.18 | 127,074.43 |
| Total | | | | | 326,330.88 |

EST is required. In addition, a minimal budget per year is necessary for a loaner or rental costs during downtime for this group of devices.

The biggest class group is Class 2, where 4,107 devices are grouped between 1 to 27 years of age. A total of 83.56% of PPM is scheduled annually, and the remaining are on a bi-annual basis, including 44 types of critical medical devices. Based on the finding of this study, the PPM schedule in class 2 is suggested to remain due to its criticality and risk of failure identified within 3 years of purchase. The devices are expected to fail at any time after being purchased; hence contingency plans or rental devices shall be planned for better service delivery to patients. Higher priority in budget allocation for maintenance and replacement is recommended for class 2 compared to class 1 and class 3. An existing warranty provided by the manufacturer or authorized party is typically within 1 or 2 years after purchase. Therefore, remedial action, such as improving the warranty service to 3 years after purchase, shall accommodate this group's critical needs. Besides, the cost saving can be maximized by purchasing a 3-year warranty, including breakdown. Hence, zero cost is required under the service contract for the first 3 years, with all PPM and breakdown costs embedded under 3 years warranty. The next group is class 3, with devices likely to fail more than 3 years after purchase, with a lesser risk of failure and complexity than class 2. A percentage of 89.52% of the PPM schedule is planned annually, and 10.48% for bi-annually. PPM frequency is suggested to be reduced bi-annually to annually with an estimated cost-saving equal to MYR 127,074.43 per year, as shown in Table 6. Like class 2, zero cost is required under the service contract for the first 3 years for class 3 devices if PPM cost is embedded under 3 years warranty. The advantage of this alternative is that the service contract fee will only start in 4th year after class 2 and class 3 devices are purchased. This framework will lead to more cost savings by reducing service contract values.

In addition to the cost-saving demonstrated in Table 6, a breakeven analysis graph can be observed in Fig. 5. This graph shows a case study analysis for a mammographic unit with 21 years of service in the class 3 category. The analysis from the graph depicts that the proposed model has a lesser service contract cost per year compared to the current maintenance practice. With the purchase cost of MYR 422,026.00 allocated by the Government for mammographic equipment, MYR 48,744.00 per year is spent on maintenance services under the present contract. However, if the proposed model is implemented, MYR 41,432.20 per year will be used, which is a 15% savings from the actual cost spent under the present contract. The service contract cost is equivalent to the

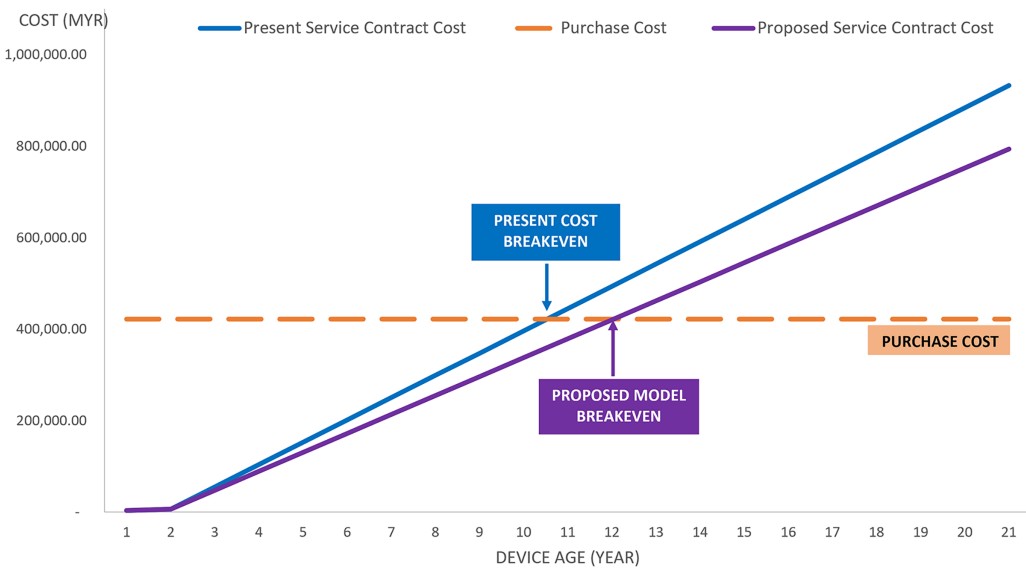

**Figure 5 Cost-saving case study for mammographic equipment.**

purchased cost in the 11[th] year for present practice and the 12[th] year for the proposed model. Therefore, the Government is suggested to replace the equipment no later than 12[th] year for better profit management. At the 12[th] year, a new device shall be purchased, and unnecessary service contract costs shall be eliminated from 12[th] to the 21[st] year of service as per the current implementation.

Another strategy to consider is the replacement plan for faulty devices. This study proposes class 2 devices be prioritized for a replacement plan, with 10% of this group's devices being more than 20 years in service, as tabulated in Table 7. A total of 57.19% of class 2 devices have more than 10 years in service, with a high number of failures throughout their lifespan. Due to the aging factor and a significant number of failure events, replacement with new units should be considered to reduce maintenance costs as the age arises. Class 3 has a similar scenario with 65.18% of devices with more than 10 years in service and should be considered for a replacement right after class 2.

## Research contributions

The development of IoT, cloud computing, and AI are continually evolving toward smart healthcare and smart city. The proposed predictive model using AI for medical devices failure prediction is categorized into classes: class 1, 2, and 3. The model accuracy is evaluated by examining other elements in performance evaluation, such as recall, precision, specificity, F1 score, and AUC. The research gap is improved by discovering new scientific findings;

- The proposed model includes 44 types of critical medical devices used in five hospital categories, with 15 healthcare facilities involved, including all critical devices used in the clinical area for patients. The data applied during the training stage includes active

**Table 7 Number of failures and ages for classes 1, 2 and 3.**

| Class | Description | No. of failures in a range | No. of devices | | | Recommendations of this study |
|---|---|---|---|---|---|---|
| | | | ≥20 years | ≥10–19 years | ≤9 years | |
| 1 | Device unlikely to fail within the first three years of purchase | 0 failures | 32 | 157 | 2,042 | • PPM frequency is suggested to be reduced from bi-annually to annually.<br>• Reducing the fee rate in the service contract to reduce operational costs.<br>• PPM can be replaced with routine inspection for small devices such as aspirators where the maintenance tasks are minimal.<br>• A yearly budget allocation for this group can be reduced for the coming year.<br>• Maintenance can be executed in-house where only average maintenance complexity with EST is required.<br>• A minimal annual budget is necessary for a loaner or rental costs during downtime. |
| 2 | Device failure within three years from purchase date | 1–10 failures<br>11–30 failures<br>≥31 failures | 126<br>173<br>115 | 716<br>698<br>521 | 1,160<br>495<br>103 | • Devices are expected to fail anytime right after being purchased. PPM schedule is suggested to remain due to its criticality. Another option is to purchase devices with three years warranty; zero cost is required under the service contract if all PPM and breakdown costs are embedded under three years warranty. The service contract fee starts in the 4th year.<br>• Contingency plans or rental devices shall be planned for better patient service delivery.<br>• Higher priority in budget allocation for maintenance and replacement is recommended for class 2 compared to other classes.<br>• Higher demand for loaner replacement and standby units.<br>• A remedial action such as improving the warranty service to three years after service shall accommodate the critical needs of this group. |
| 3 | Device failure after three years from purchase date | 1–10 failures<br>11–30 failures<br>≥31 failures | 292<br>151<br>29 | 658<br>131<br>14 | 644<br>35<br>2 | • A lesser risk of failure and complexity compared to class 2.<br>• PPM frequency is suggested to be reduced from bi-annually to annually.<br>• Another option is to purchase devices with three years warranty; zero cost is required under the service contract if PPM cost is embedded under three years warranty. The service contract fee starts in the 4th year.<br>• Moderate priority in budget allocation compared to class 2.<br>• Moderate demand for loaner replacement and standby units.<br>• Moderate allocation for budget allocation yearly. |
| **Total** | | | 918 | 2,895 | 4,481 | |

medical devices with data on maintenance cost, devices with more than 20 years in service, and sample size up to 8,294 devices resulting in the highest accuracy of 79.50%. Only eight out of 17 features are significant after sensitivity analysis, with a reduction of training time from 11.49 to 7.908 min. A robust model to predict failure in three classes using AI is introduced with fewer features, shorter training time, and comprehensive cost analysis.

- This proposed model can forecast the first occurrence of failure in classes 1, 2, and 3 after medical device is purchased for comprehensive maintenance planning and budget utilization which is currently not in the field of study. The maintenance shall be arranged before the first failure event. Class 1 is the less critical device with zero failure. Class 2 is identified as the most crucial and should be attended to, with the first failure occurrence likely to be within 3 years after purchase. Class 3 is at medium risk; devices are likely to fail after 3 years. To reduce the likelihood of future failures, device replacement for class 2 is in higher priority, followed by class 3. A replacement is proposed by a number of failures and year of service category. This is crucial for any country during crisis management, such as COVID-19 pandemic, where excellent and reliable equipment is highly utilized.

- This article compares the country's role in organizing budgets while reducing costs in maintenance management. A new PPM schedule and replacement plan frequency are proposed and strategized based on actual needs. Comprehensive strategic management by criticality and devices maintenance history using AI improves maintenance and operational cost. Replacement strategy shall be executed to class 2, then to class 3, depending on the age of the devices and the likelihood of failures.

- Two different techniques between ML and DL were compared, and ML has better performance in accuracy, recall, precision, specificity, and F1 Score. DL has the advantage of shorter training time; however, the accuracy is lower than the ML technique. The accuracy shall be further improved in future work by introducing unstructured maintenance notes written by technical personnel after rectification work is completed.

## CONCLUSIONS

A robust predictive model for 44 types of critical medical devices is proposed in this article for smart healthcare management within three failure classes; class 1, 2, and 3. Class 1 includes devices unlikely to fail within the first 3 years from the purchase date; class 2 is the devices that are likely to fail within 3 years from the purchase date, and class 3 is the devices that are likely to fail more than 3 years after purchase. The result concludes Ensemble Classifiers have better performance than SGDM optimizer and attained the highest accuracy of 79.50% with the highest recall, specificity, and F1 score values after significant features are recognized. Replacement of class 2 devices is expected to improve critical medical devices' uptime and optimize yearly budget allocation. An aging medical device with a high number of failures is impacted by high maintenance costs, higher downtime, and exceeding its lifespan, which can jeopardize patient safety. The lifespan of medical

devices is also affected by various factors such as age, utilization, environment, user handling, specification, availability of spare parts, safety, *etc.* Although PPM frequency is suggested to be reduced in classes 1 and 3, technical personnel shall strengthen PPM tasks to accommodate the needs before reaching the following PPM schedule and ensure useful life is not compromised. The functionality and safety should be guaranteed during PPM as per IEC60601 requirements. In addition, the predictive model can be used as a reference for new medical device procurement. AI's capability to understand hidden patterns of historical device data can facilitate decision-making by making the replacement process faster. This can reduce time spent on data analysis, where the process optimization tasks can be done automatically without human intervention. In addition, the predictive maintenance capability achieved by AI allows healthcare institutions to maintain industrial medical equipment based on the times and conditions of operation, allowing the equipment to increase its performance and life cycle. Having more information in a structured way allows clinical engineers in charge to make decisions faster and more efficiently. In addition, we have demonstrated the feasibility of leveraging AI technology in developing smart maintenance systems. This contribution has shown promising results to be implemented in our healthcare facilities by integrating existing asset management systems and the developed AI predictive model. We have also presented how government hospitals can save their maintenance budget by having a smart maintenance system using AI predictive model. The limitation of this research is that data collection is based on existing data in ASIS, and there are possibilities where no failure event is recorded even though it has occurred. Other factors, such as user disregard to launch a failure complaint, error during human intervention with the system, or any system issues, might also be the constraint. Future work shall include adding an unstructured maintenance note during failure events to improve the model's accuracy and reliability.

## ACKNOWLEDGEMENTS

We would like to express our highest appreciation to the Director General of Health Malaysia for the medical device dataset of healthcare facilities in Peninsular Malaysia.

### Funding

The authors received no funding for this work.

### Competing Interests

The authors declare that they have no competing interests.

### Author Contributions

- Noorul Husna Abd Rahman conceived and designed the experiments, performed the experiments, analyzed the data, performed the computation work, prepared figures and/or tables, authored or reviewed drafts of the article, and approved the final draft.

- Muhammad Hazim Mohamad Zaki performed the experiments, performed the computation work, prepared figures and/or tables, and approved the final draft.
- Khairunnisa Hasikin conceived and designed the experiments, performed the experiments, analyzed the data, performed the computation work, prepared figures and/or tables, authored or reviewed drafts of the article, and approved the final draft.
- Nasrul Anuar Abd Razak conceived and designed the experiments, performed the experiments, analyzed the data, performed the computation work, authored or reviewed drafts of the article, and approved the final draft.
- Ayman Khaleel Ibrahim conceived and designed the experiments, performed the experiments, analyzed the data, performed the computation work, authored or reviewed drafts of the article, and approved the final draft.
- Khin Wee Lai conceived and designed the experiments, performed the experiments, authored or reviewed drafts of the article, and approved the final draft.

## Data Availability

The matlab code and raw data are available in the Supplemental Files.

## Supplemental Information

Supplemental information for this article can be found online at http://dx.doi.org/10.7717/peerj-cs.1279#supplemental-information.

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
