# Peer review of "Predicting medical device failure: a promise to reduce healthcare facilities cost through smart healthcare management"

_PeerJ Computer Science, doi:10.7717/peerj-cs.1279_

## Round 0.1 · original submission · Major Revisions

Two reviewers suggested revisions to this manuscript. Please carefully revise the manuscript and resubmit for a second review.

·

Basic reporting

2- In the summary, the database is explained in the method section, but the deep learning methods used are not mentioned. The names of the methods should be given in this section.
3- Reference error exists on lines 289, 317, 364, etc. "Error! Reference source not found." terms should be corrected. (in pdf version)
4- BER is used for Bit Error Rate. The usage of BER for Beyond Economic Repair, may pose misunderstandings. Use another abbreviation for this (like BeEcR, BEcR or etc.)

Experimental design

1- The most important point that intrigues me in this study is the following. It is seen that the sample numbers of the classes are not equal in the database. What kind of basis do you have as to whether this situation has an effect on the result? Can you suggest a solution for this problem?

Validity of the findings

1- The most important point that intrigues me in this study is the following. It is seen that the sample numbers of the classes are not equal in the database. What kind of basis do you have as to whether this situation has an effect on the result? Can you suggest a solution for this problem?
5- In order for the values given in Table 6 to be more meaningful, it should be added how much profit is obtained by proportioning these devices to the average annual maintenance cost.

Additional comments

Predicting medical device failure: A promise to reduce healthcare facilities cost through smart healthcare management

In this study, the estimation of the failure period of medical devices was made with machine learning and deep learning methods. The effect of this estimation on the maintenance cost has been examined. I believe that clarifying the following points in this study will increase the clarity and fluency of the article.

1- The most important point that intrigues me in this study is the following. It is seen that the sample numbers of the classes are not equal in the database. What kind of basis do you have as to whether this situation has an effect on the result? Can you suggest a solution for this problem?
2- In the summary, the database is explained in the method section, but the deep learning methods used are not mentioned. The names of the methods should be given in this section.
3- Reference error exists on lines 289, 317, 364, etc. "Error! Reference source not found." terms should be corrected. (in pdf version)
4- BER is used for Bit Error Rate. The usage of BER for Beyond Economic Repair, may pose misunderstandings. Use another abbreviation for this (like BeEcR, BEcR or etc.)
5- In order for the values given in Table 6 to be more meaningful, it should be added how much profit is obtained by proportioning these devices to the average annual maintenance cost.

·

Basic reporting

The authors made a study to determine the most suitable model among machine learning and deep learning techniques to use in smart health services. The article is generally very beautifully designed. However, if the following corrections are considered, I think the content of the article may be more beautiful.

Line 35: There is no need for a comma when typing 8,294. It is sufficient to write 8294. For numbers with more than 4 digits, thousands of separators should be used.
Line 44-46: “An optimized predictive machine learning model based on Ensemble Classiûer produces 77.90%, 87.60 %, and 75.39% for accuracy, specificity, and precision compared to 70.30%, 83.71%, and 67.15% in deep learning model or SGDM optimizer, respectively.” Because the sentence is long, it creates confusion of meaning.
Line 78: USD145 or USD 45.2 : One has a space, one does not. It should have a standard usage.
Line 71: There is no space before some parentheses.
There are typos throughout the article. Comma, period, parenthesis, space…..
Line 54: “MYR326,330.88” what are these abbreviations? It should be explained where it is used for the first time. In addition, it should be decided whether to write adjacent or separate. It should be standard in the entire article.
Line 108: Is it correct to use “(Hilmi et al. 2021), (Mahfoud et al. 2018)”? (Hilmi et al. 2021; Mahfoud et al. 2018)" seems to be used in this way in the literature?
Line 152: The initials of “information technology (IT)” should be capitalized.
Line 289: In text “Error! There is a warning message like "Reference source not found". Available in many places.
Line 301: The formula is given, but it is not clear what the variables in the formula mean.
Line 202: AUC expansion should be given where it is first used.
In addition, abbreviations, formulas and terms should be written very carefully while writing the article.

Experimental design

no comment

Validity of the findings

no comment

Additional comments

A comprehensive literature review was conducted in the review study. There are semantic gaps in the narrative of the studies in the literature and these should be reconsidered. There are numerous typos throughout the article and they should be corrected.

---

## Round 0.2 · Minor Revisions

According to the reviewers' comments, please revise your manuscript and resubmit.

·

Basic reporting

no comment

Experimental design

no comment

Validity of the findings

no comment

Additional comments

The changes made in line with the reviews made the study more understandable and interesting to the reader. The study is suitable for publication as it is.

·

Basic reporting

Table 2 resolution is low. It was taken from the program and put directly into the article. The texts on the axes are not readable.

Check for language.

Experimental design

A very well constructed article. I believe that their contribution to the literature will be at a high level.

Validity of the findings

They have tried many methods to find the highest success. The findings are quite high. It will shed light on future studies on this subject.

---

## Round 0.3 · accepted · Accept

The authors have addressed all of the reviewers' comments. This manuscript is ready for publication.

·

Basic reporting

The changes made in line with the reviews made the study more understandable and interesting to the reader. The study is suitable for publication as it is.

Experimental design

Everything ok.

Validity of the findings

Everything ok.

Additional comments

The changes made in line with the reviews made the study more understandable and interesting to the reader. The study is suitable for publication as it is.

·

Basic reporting

Necessary corrections have been made in the article. It is suitable for publication.

Experimental design

Necessary corrections have been made in the article. It is suitable for publication.

Validity of the findings

Necessary corrections have been made in the article. It is suitable for publication.

Additional comments

Necessary corrections have been made in the article. It is suitable for publication.